# Human mobility and malaria risk in peri-urban and rural communities in the Peruvian Amazon

Joaquin Gomez[1], Alessandro Grosso[2], Mitchel Guzman-Guzman[1], Stefano Garcia Castillo[1¤], Marcia C. Castro[3], Katherine Torres[1,4], Joseph M. Vinetz[1,5]*, Dionicia Gamboa[1,6]*

1 Laboratorio ICEMR- Enfermedades Emergentes, Laboratorios de Investigación y Desarrollo, Facultad de Ciencias e Ingeniería, Universidad Peruana Cayetano Heredia, Lima, Perú, 2 Global Health Institute, Department of Family Medicine and Population Health, University of Antwerp, Antwerp, Belgium, 3 Department of Global Health and Population, Harvard T.H. Chan School of Public Health, Boston, Maryland, United States of America, 4 Laboratorio de Malaria, Laboratorios de Investigación y Desarrollo, Facultad de Ciencias e Ingeniería, Universidad Peruana Cayetano Heredia, Lima, Perú, 5 Section of Infectious Diseases, Department of Internal Medicine, Yale School of Medicine, New Haven, Connecticut, United States of America, 6 Laboratorio de Malaria: Parásitos y vectores, Laboratorios de Investigación y Desarrollo, Facultad de Ciencias e Ingeniería, Universidad Peruana Cayetano Heredia, Lima, Perú

¤ Current address: Department of Molecular Microbiology and Immunology, Johns Hopkins Bloomberg School of Public Health, Baltimore, Maryland, United States of America
* joseph.vinetz@yale.edu (JMV); dionicia.gamboa@upch.pe (DG)

**Data Availability Statement:** Data available at https://clinepidb.org/ under the name "Amazonia ICEMR Peru Cohort."

## Abstract

### Background

While the global burden of malaria cases has decreased over the last two decades, the disease remains a major international threat, even on the rise in many regions. More than 85% of Peruvian malaria cases are in the Amazonian region of Loreto. Internal mobility primarily related to occupation is thought to be primarily responsible for maintaining endemicity and introducing and reintroducing malaria parasites into areas of anophelism, a challenge for malaria eradication. This study focuses on identifying the sources of malaria transmission and patterns of human mobility in order to understand the movement and transmission of the parasite.

### Methods

The assessment of connectivity produced by human mobility was evaluated in three districts of Loreto, through 10 cross-sectional population screening from 2018 to 2020. We used social network analysis (SNA) to obtain weighted and unweighted degrees of connectivity and explore its variability by socio-demographic characteristics. In addition, we integrated travel history and malaria incidence data to estimate parasite connectivity due to internal human mobility between locations. Finally, we used logistic multivariate regressions to explore the factors associated with *Plasmodium spp.* infection in mobile individuals.

### Results

We found that internal human mobility results in high connectivity between communities from the Mazan, Iquitos, and San Juan Bautista districts. We identified nearby destinations

**Funding:** The Amazonian International Center of Excellence in Malaria Research received funding for this work through the Cooperative Agreement U19AI089681 from the US Public Health Service, National Institutes of Health/National Institute of Allergy and Infectious Diseases, USA to JMV, and from a FOGARTY global infectious disease training grant (2D43TW007120-11A1, NIH-USA to JMV). The funders had no role in study design, data collection and analysis, decision to publish, or preparation of the manuscript.

**Competing interests:** The authors have declared that no competing interests exist.

that may act as sinks or sources for malaria transmission, including densely populated towns and rural campsites. In addition, we found that being a male, traveling to rural camp-sites, and working outdoors are associated with *Plasmodium spp.* infection in travelers from the Mazan district.

## Conclusions

We provide compelling evidence about how human mobility connects rural communities in the Peruvian Amazon. Using SNA, we uncovered district-specific patterns and destinations, providing further evidence of human mobility heterogeneity in the region. To address the challenge of human mobility and malaria in this setting, geographic heterogeneity of malaria transmission must be considered.

## Author summary

Malaria transmission is complex, involving interactions of parasite, vector biology and ecology, human immune response, and human host behavior. We used social network analysis to understand how the daily lives including occupation-related mobility leads to the maintenance of malaria endemicity and transmission. Rural communities (within the districts of Mazan) and peri-urban communities around the city of Iquitos have different socio-demographic characteristics that determine malaria transmission rates. Networks of fixed rural communities and occupation-related distant camp sites play a key role as reservoirs and sources of parasite movement. Factors including male sex, outdoor occupational activity outdoors, and having rural camp sites as travel destination increase the risk of *Plasmodium* infection. Factors associated with occupation-related mobility are correlated with incidence and prevalence of malarial disease and parasite infection rates, respectively. This work provides a detailed understanding of the importance of work-related travel to target in malaria elimination programs, key to maintaining malaria transmission both in the Amazon region and more generally.

## Introduction

Malaria remains a global public health burden worldwide, with 249 million cases and more than 608,000 deaths reported only in 2022 [1]. In 2022, in the Americas, the estimated annual number of malaria cases was 552,000, with *Plasmodium vivax* representing 72% of the total burden [1]. National malaria control programs (NMCPs) in the Americas have been effective in many places, resulting in malaria elimination in countries such as Argentina, Belize, El Salvador and Paraguay [1]. However, malaria control, and its eventual elimination, poses a difficult challenge for countries within the Amazon basin, resulting in heterogeneous progress against the disease. Human mobility—that is, the movement of people (whether for work or other reasons) who move malaria parasites among regions with anophelism [2]—has increasingly become recognized in the Amazon region as a key challenge that must be addressed to achieve malaria elimination [3]. This concept is globally applicable outside of Amazonia [2,4,5].

Understanding human mobility is essential in the fight against malaria transmission, as it facilitates the introduction of parasites into malaria-free areas, leading to the re-establishment

of the disease [1–13]. For example, epidemiological studies in Africa and Brazil attribute the failure of control plans and the persistence of the disease to the importation of parasites and the high connectivity resulting from commuting and migration between areas with heterogeneous transmission [3–5]. Human mobility has been shown to contribute to the spread of drug-resistant *Plasmodium* into vulnerable regions [6,7].

Addressing human mobility is a complex challenge but it is important for the long-term goal of malaria eradication [8–10]. In Peru, the Loreto region bears the highest malaria burden, with 22,698 reported cases in 2022, constituting 84% of the total cases in the country [11]. Even though malaria transmission in the Peruvian Amazon reached its lowest levels over the past several years [11], conditions continue that could lead to sudden reeemergence as happened in the mid-1990s [2].

Malaria in the Amazon region is typified by transmission in riverine communities where conditions for the primary vector, *Nyssorhynchus darlingi*, enable this highly anthropophilic mosquito species to flourish [6–13]. Previous studies suggest that inhabitants of these communities may represent a highly mobile population, recording daily movements, many lasting more than one day and covering distances exceeding 10 km [12,13]. Given the micro-heterogeneity of malaria transmission in the region [14], human movement, particularly related to economic activity, leads to high exposure to malaria vectors during trips to areas with high local transmission, leading to importation and subsequent transmission upon their return to their community of origin (14).

This study tested the hypothesis that identifying malaria parasite transmission sources, and determining within country, inter-regional human mobility patterns is important and underlies parasite strain movement. Understanding the role of human mobility of *Plasmodium* is key to developing successful new malaria control and elimination strategies.

## Methods

### Ethics statement

Data used in this study was collected by two projects within the Amazonian International Center of Excellence for Malaria Research (ICEMR) that were reviewed and approved by the Institutional Ethics Committee in Research (CIEI) from Universidad Peruana Cayetano Heredia in Lima, Peru (SIDISI code: 101518 and 101497). All adult participants (18 years and older) signed informed consent before the study enrollment and blood sample collection. Younger than 18 years individuals provided an informed assent, and their parents or guardians signed an informed consent before enrollment.

### Study area

Data used in this study were obtained from inhabitants of rural communities from Mazan, Iquitos and San Juan districts (Fig 1) of Maynas province in Loreto region. The participants were recruited from 2018 to 2020 as part of population-based malaria projects. Communities were selected based on their malaria risk and established transportation routes to ensured sample preservation. Overall, Loreto is characterized by a tropical climate with two marked seasons. The rainier season is from November to May, with a minimum temperature of 17˚C and an average humidity of ~80%. The less wet season is from June to October with temperatures around 36˚C [15]. Malaria transmission in the region is highly localized, dominated by *Plasmodium vivax* transmitted by *Ny. darlingi* [16,17].

The Mazan district (North of Iquitos city) comprises rural riverine communities scattered throughout the Mazan and the Napo rivers basin. Near the confluence of both rivers lies the district's capital, Mazan town (3.503˚ S, 73.094˚ W), with approximately 3800 inhabitants,

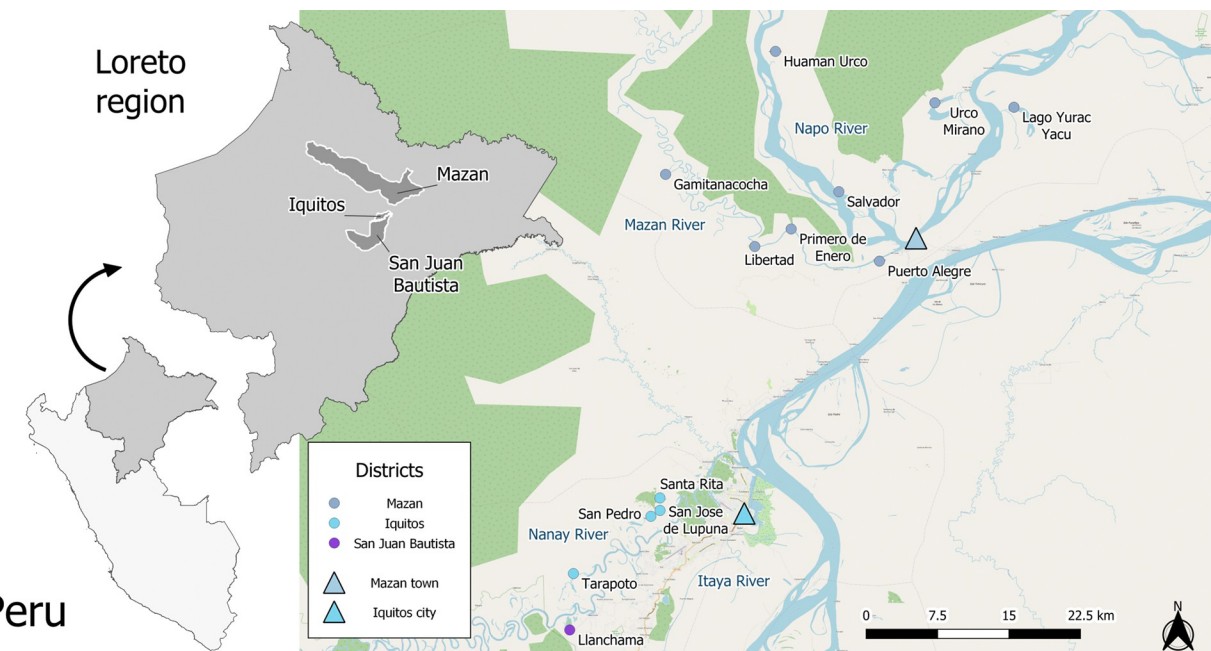

**Fig 1. Map of the study area.** In sky blue color, communities from the Iquitos district; in purple, from the San Juan district; in ice blue, from the Mazan district. Important urban settlements are shown as triangles. In ice blue, the Mazan town; in sky blue, the Iquitos city. The map was built using QGIS 3.16 (QGIS Development Team, 2023. QGIS Geographic Information System. Open-Source Geospatial Foundation Project. http://www.qgis.org/). We used OpenStreetMap (http://www.openstreetmap.org/) layers licensed under the Open Data Commons Open Database License (ODbL). Shapefiles can be freely accessed on the Peruvian National Institute of Statistics and Informatics (INEI, https://ide.inei.gob.pe/) website.

almost one-third of the district's total population. Residents from the Mazan district are mainly "mestizos" (i.e. Peruvian citizens without a specific ethnic group), and rural communities rely heavily on agricultural and extractive activities (fishing, hunting, and logging). Malaria transmission has been diminishing in the past few years, the annual parasite index (API) is ~ 60 cases per 1000 inhabitants [18] and riverine communities (> two-thirds of the district's population) are still at higher risk of acquiring malarial infection compared to Mazan town.

The San Juan district (south of Iquitos city) consists of urban, peri-urban, and rural communities scattered along the Iquitos-Nauta road or along the Itaya river and Nanay river (16). Although malaria transmission is low (2022 API, 2.75 cases per 1000 inhabitants) [18], residents from peri-urban and rural communities have higher malaria risk attributed to poor housing conditions, lack of essential services including access to efficient malaria diagnosis, and presence of the vector [6,15,16].

Iquitos district comprises urban (within the Iquitos city), peri-urban, and rural communities dispersed along the Nanay River [19]. The data used for this study came from three communities from a rural riverine site known as Lupuna. Inhabitants from Lupuna are mainly mestizo who work in agricultural activities (charcoal production and cassava cultivation), fishing (Nanay's riverbank), and occasional hunting. Malaria transmission within Iquitos is low (2022 API: 5.23 cases per 1000 inhabitants) [18], compared to rural areas outside the city such as Lupuna. In contrast to the city of Iquitos, Lupuna inhabitants living near forested areas and along the rivers are at higher risk of exposure to malaria vectors.

Malaria transmission in Loreto region is mainly driven by the *Ny. darlingi* vector, since its emergence around Iquitos in the 1990s [20], although other species have recently reported in Datem del Marañón province [21]. The abundance of *Ny. darlingi* varies between districts and communities, even within the same community. Heterogeneous vector distribution and

abundance in communities from Mazan river and Napo river has been reported previously [22] and recently it was also confirmed the highest malaria transmission outdoors during rainy season, when there is an increase on the breeding sites in communities adjacent to the highway or along the Nanay river [23].

### Study design

Data from inhabitants of communities within the Iquitos, San Juan, and Mazan districts were collected by the P3 project (SIDISI code 101497). This project focused on obtaining sociodemographic data and blood samples from adult (>18 years old) inhabitants in communities with high malaria transmission (high malaria positivity index reported by the Regional Direction of Health from Loreto within eight weeks before the screening) [24] after informed consent was signed. Eight cross-sectional population screenings were performed during 2018 to 2020, and qPCR diagnosis was only available for adult participants (S1 Table).

The P1 project, complementary to P3 (SIDISI code 101518), collected sociodemographic data and blood samples from residents of seven communities within the Mazan district (S1 Table). This project addressed the factors and conditions associated with residual malaria transmission in rural riverine and hard-to-reach Amazon populations. Therefore, two whole population cross-sectional surveys were deployed in July and October 2018, and qPCR diagnosis was available for every participant (S1 Table). Data collection (demographics and blood samples) was conducted on participants 3 months of age and older who lived in the selected communities and were willing to provide informed consent (adults) or informed assent (<18 years old) with the approval of their caregivers.

Participants were excluded from both projects if they disagreed with the sampling procedures or if the field physician observed an acute or chronic condition that might affect their ability to provide informed consent.

Sample size estimates were not conducted as each screening aimed to enroll the entire target population. A census was conducted two weeks before cross-sectional screenings to map households and estimate the total population. Data collectors visited the mapped households during the first screening in each community to present the project and invited eligible residents to participate in the study. If participants were absent during the first visit, data collectors revisited the household within the next two days to ensure enrollment. Close communication was maintained with local leaders to provide information about newly established households between screenings. House members were invited to participate in the project when a new household joined the community.

During these censuses, a short Open Data Kit (ODK) questionnaire and a handheld Global Positioning System (GPS) device (Garmin's GPSMAP 60CSx, Garmin International Inc., USA) were used to geo-reference households. During cross-sectional screenings, sociodemographic data were collected employing two data collection electronic data capture (EDC) tools at two different times. From 2018 to mid-2019, the P3 project designed surveys through the ODK aggregate online tool and collected in-field data using the ODK-collect mobile application. On the other hand, from mid-2019 to 2020 and during the 2018 cross-sectional whole population surveys, the P3 and P1 projects designed surveys and collected in-field data using the REDcap online (http://www.projectredcap.org/) and REDcap mobile applications, respectively. Data collection instruments were written in Spanish. The questionnaires were piloted in collaboration with local researchers from Loreto and nearby communities who assessed their clarity and ease of understanding. Several rounds of testing were conducted to ensure that the instruments were error-free.

Cross-sectional surveys at any time and for any project had similar structures and questions. First, enrolled participants filled out an individual-level questionnaire (age, gender, socioeconomic, recreational, occupational activities, and travel-related questions), and blood samples were taken by finger-prick and stored in filter papers (2018 to 2019—P3) or EDTA-embedded microtainers (2018—P1, mid-2019 to 2020—P3). Later, the head of the family filled out a house-level questionnaire (house structural characteristics and services available). Finally, blood samples were used for in-field malaria microscopy diagnosis (thick and thin blood smears) and stored for qPCR laboratory procedures. Although the enrollment criteria were the same (except for the age), the sample processing differed. The P1 project performed microscopy and qPCR on all available samples, while P3 focused on adult blood samples. Therefore, qPCR diagnosis was available for all participants in P1 but only for adults in P3.

## Laboratory procedures

**Microscopy.** Blood samples were diagnosed by microscopy in the field on the day of collection. Thick and thin blood smears were stained with 10% Giemsa solution for 10 minutes, and parasite density was determined by the number of asexual parasites by 200 leucocytes (L), assuming a concentration of 8000 L/uL of blood. Samples were classified as negative if no parasites were found within 100 fields of microscopy [25]. Later, a second diagnosis round was performed by a microscopy expert at the Iquitos city laboratory, and a final quality control (all positive slides and 10% of negatives) was done in Lima, Peru.

**Real-Time Quantitative PCR (qPCR).** DNA was extracted and purified from whole blood samples or filter paper employing the E.Z.N.A. Blood DNA mini-Kit (Omega Bio-Tek, Georgia, USA) following manufacturer instructions with minor modifications (addition of TEN buffer [20 mM Tris-HCl, pH 8.0; 2 mM EDTA, pH 8.0; 0.2 M NaCl] supplemented with SDS 10% w/v). Purified DNA was stored at 4˚C for immediate use or at -20˚C for downstream analysis. Quantitative polymerase chain reaction (qPCR) testing to detect malaria species (*Plasmodium spp*.) was done following the guidelines from Mangold *et al.* with slight modifications, including the use of PerfeCTa SYBR Green FastMix (Quanta Biosciences, MD, USA) [26]. Later, positive samples were subjected to a second qPCR round using TaqMan probes to detect species-specific (*P. vivax* and *P. falciparum)* rRNA from the 18S subunit [27].

**Data analysis.** Given the proximity of Llanchama to the Iquitos district and the low sample size in San Juan, this community was considered part of Iquitos. Consequently, collected data were classified into two groups according to the district of origin (Table 1). Databases were managed and analyzed using the R statistical language (*V. 4.1.2*) and the RStudio software (*RStudio*: *Integrated Development Environment for R. RStudio, PBC, Boston, MA* http://www. rstudio.com/). Baseline characteristics between districts were compared using the Fisher's exact test and the Chi-squared test. The age of the population was categorized into four groups (Table 2). Previous reports show that inhabitants from 15 years old engage in economic activities [28,29]. By categorizing the age from 15 to 40, we include young inhabitants who may be conducting economic activities in the same group as young adults.

Social network analysis (SNA) was used to explore the connectivity produced by human mobility, considering communities as nodes and travel events as edges. In SNA, the degree is defined as the number of interactions a node has. In our study, nodes are represented by communities and the degree is the number of recorded travels–regardless of the direction–between two communities. SNA can also assess directionality, dividing the degree into the out-degree and in-degree. In our study, the out-degree represents the number of outgoing trips from a given community to other destinations. The in-degree represents the number of incoming trips from other communities. A community with inhabitants traveling to other destinations

**Table 1. Overall participants and mobile individual collection summary.**

| Community | Households | Individuals | Records | Travel records | |
|---|---|---|---|---|---|
| | n | n | N | N | % |
| **Iquitos** | | | | | |
| Llanchama* | 55 | 113 | 113 | 46 | 40.7 |
| Lupuna | 94 | 205 | 333 | 63 | 18.9 |
| San Pedro | 46 | 176 | 177 | 49 | 27.7 |
| Santa Rita | 98 | 385 | 495 | 201 | 40.6 |
| Tarapoto | 35 | 60 | 60 | 37 | 61. |
| **Total** | **328** | **939** | **1,178** | **396** | **33.6** |
| **Mazan** | | | | | |
| Gamitanacocha | 21 | 86 | 125 | 47 | 37.6 |
| Huaman Urco | 56 | 73 | 228 | 31 | 13.6 |
| Lago Yuracyacu | 61 | 187 | 358 | 72 | 20.1 |
| Libertad | 101 | 288 | 519 | 139 | 26.8 |
| Primero de Enero | 42 | 113 | 186 | 46 | 24.7 |
| Puerto Alegre | 89 | 258 | 449 | 128 | 28.5 |
| Salvador | 163 | 472 | 981 | 128 | 13.1 |
| Urco Mirano | 132 | 596 | 966 | 137 | 14.2 |
| **Total** | **665** | **2,073** | **3,812** | **728** | **19.1** |

* Llanchama belongs to the San Juan Bautista district; however, we considered as part of Iquitos in regression analysis

will show a high out-degree, while a community receiving travelers from other destinations will have a high in-degree. In addition, the degrees can be expressed as non-weighted (without considering the number of trips conducted to each destination) and weighted [30,31]. We calculated several SNA metrics (degree, in-degree, out-degree, and their weighted versions) to identify relevant nodes within the network. Moreover, for assessing differential connectivity patterns, we extended this analysis to identify heterogeneities in mobility patterns by age groups, sex, and travel reasons between districts, following the study by Pindolia *et al* [32].

Parasite connectivity was assessed by calculating *relative parasite importation* score as proposed by Tessema *et al* [33]. We used verbally-obtained travel history data (time of stay and destination) to assess the aggregated time at risk between pairs of locations. Later, we adjusted aggregated time at risk by the ratio of the mean incidence between the origin and destination. The analysis was performed at the district (only with pairwise, between-district travels) and the community levels (all travels). District-level mean incidence from 2018 to 2020 was calculated based on the Iquitos Regional Health Direction (DIRESA) reports. Community-level incidence for our listed locations (i.e., screened communities and destinations reported by participants) was estimated using reports from the same institution but matching the names between our database and theirs based on string similarity. As names entered by data collectors and those used by the DIRESA were not identical, we used a string comparison algorithm that filtered reported cases from communities with a string similarity Jaro-Wrinkler index of at least 0.1 to our listed locations [34]. As communities in different districts with the same name are common in Loreto, we used district-of-origin data from DIRESA to ensure that the Annual Parasite Index (API) calculation was appropriate. Noteworthy, the score proposed by Tessema *et al*. evaluates the role of locations as sources rather than sinks [33].

The role of travel characteristics on *Plasmodium spp.* infection in travelers from Iquitos and Mazan districts was evaluated by building a Generalized Linear Model. Specifically, we assessed the effect of overnight trips, travel reason, travel destination, number of trips in the

last month, and other sociodemographic variables on the odds of positive *P. vivax or P. falciparum* qPCR results. Variables were pre-selected using univariate analysis and a threshold p-value of 0.2. Afterward, we built the multivariate model using a forward selection process, adding pre-selected variables one by one and keeping only those with a p-value of at least 0.2. The correlation of independent variables was assessed using the *variance inflation factor* (VIF), and highly correlated variables were dropped from the model. Finally, odds ratios (OR) and adjusted odds ratios (AOR) were estimated for the univariate and adjusted multivariate models.

## Results

### Sociodemographic characteristics of the general and mobile populations

We collected data from 993 households and 3,012 individuals (4,l90 records considering repeated measures) between 2018 and 2020. Sociodemographic data were classified according to participants' district of origin: Iquitos (328 households; 939 individuals; 1,178 records) or Mazan (665 households; 2,073 individuals; 3,812 records). Regarding human mobility, 33.6% (N = 396/1178) of the surveys/records from Iquitos presented travel history in the last month, while the percentage reduced to 19.1% (N = 728/3812) in Mazan. The distribution of records by community within each district is described in Table 1.

The sociodemographic characteristics of the general population are described in Table 2. The sex distribution within the population was the same in both districts (p-value> 0.05, Table 2); however, the age distribution differed. Mazan communities showed a significantly higher proportion of younger individuals (<15 years) (p<0.05, Table 2) than those in Iquitos. In addition, education levels were significantly lower in Mazan district inhabitants, with a significantly higher proportion (80%, p<0.05, Table 2) of participants with complete primary education or lower. Working in outdoor activities (logging, hunting, farming, or fishing) was more common among Mazan dwellers, and a smaller fraction (Mazan: 14%, Iquitos: 23%, p<0.05, Table 2) of the population reported receiving an income in the last month compared to the inhabitants from Iquitos district. *Plasmodium spp.* infection was more common in Iquitos inhabitants (p<0.05, Table 2). In the last month (30 days), 3.4% of participants from Iquitos reported being diagnosed with malaria, while 19.9% and 3.1% had a positive malaria diagnosis by qPCR for *P. vivax* and *P. falciparum* at the time of the survey, respectively. On the other hand, only 2.4% of participants in the Mazan district reported being diagnosed with malaria in the last month, while 4.4% and 1.1% had a positive malaria diagnosis by qPCR for *P. vivax* and *P. falciparum* at the time of the survey, respectively.

Travel/mobility profiles different among the inhabitants of Iquitos and Mazan. Mobile participants from Mazan traveled less frequently, typically once per person in the last 30 days. In contrast, traveling more than once in the last 30 days was much more common (58%, N = 228, p<0.05) in inhabitants from Iquitos communities. The time spent at the destination was similar for mobile individuals in both districts. Most of the population stayed for hours at their destination (Iquitos: 72.70%, Mazan: 66.62%), with a slightly higher, although not significant, proportion of trips by inhabitants from Iquitos communities (Table 2). When transforming the length of stay to weeks, we found that Iquitos travelers had a moderately longer length of stay than those from the Mazan district (Iquitos: IQR = [0.0238–0.143], Mazan: IQR = [0.0179–0.143], W = 161,115, p-value <0.05). Travel motives of the mobile population differed between districts. While most travelers in the Iquitos district traveled for personal reasons (paperwork, studies, military service, or domestic violence) or recreational activities, the most common travel reasons of Mazan inhabitants were economic (work, public subsidy) and family matters (Table 3).

**Table 2. Baseline sociodemographic characteristics of the general population.**

| Characteristic | Iquitos, N = 939[1] | Mazan, N = 2073[1] | p-value[2] |
|---|---|---|---|
| **Sex** | | | 0.5 |
| Female | 472 (50%) | 1,013 (49%) | |
| Male | 467 (50%) | 1,060 (51%) | |
| **Age** | | | <0.001 |
| < 15 | 226 (24%) | 948 (46%) | |
| 15 to 40 | 342 (36%) | 586 (28%) | |
| 40 to 60 | 221 (24%) | 351 (17%) | |
| 60 or older | 150 (16%) | 188 (9.1%) | |
| **Study level** | | | <0.001 |
| None or primary | 527 (56%) | 1,393 (80%) | |
| Secondary or higher | 412 (44%) | 349 (20%) | |
| Not available | 0 | 331 | |
| **Main economic activity** | | | 0.038 |
| Indoors | 627 (67%) | 1,303 (63%) | |
| Outdoors | 312 (33%) | 770 (37%) | |
| **Income in the last month** | | | <0.001 |
| Yes | 220 (23%) | 291 (14%) | |
| No | 719 (77%) | 1,782 (86%) | |
| Not available | 0 | 2 | |
| **Malaria diagnosis in the last 30 days** | | | 0.1 |
| No diagnosis | 907 (97%) | 2,024 (98%) | |
| Positive diagnosis | 32 (3.4%) | 49 (2.4%) | |
| **Microscopy results** | | | <0.001 |
| Negative | 646 (95%) | 1,835 (99%) | |
| *P. falciparum* | 4 (0.6%) | 7 (0.4%) | |
| *P. vivax* | 32 (4.7%) | 20 (1.1%) | |
| Mixed | 1 (0.1%) | 0 (0%) | |
| Not available | 256 | 211 | |
| **qPCR results** | | | <0.001 |
| Negative | 524 (77%) | 1,760 (95%) | |
| *P. falciparum* | 21 (3.1%) | 20 (1.1%) | |
| *P. vivax* | 129 (19%) | 82 (4.4%) | |
| Mixed infection | 9 (1.3%) | 0 (0%) | |
| Not available | 256 | 211 | |
| **House type** | | | 0.6 |
| Closed | 872 (93%) | 1,915 (92%) | |
| Opened | 67 (7.1%) | 158 (7.6%) | |
| **Traveled last month** | | | |
| Yes | 337 (36%) | 501 (24%) | <0.001 |
| No | 602 (64%) | 1,572 (76%) | |

[1]n (%)

[2]Pearson's Chi-squared test; Fisher's exact test

## Social Network Analysis of the mobile sampled population

We recorded 1124 commuting movements that implied traveling out of the participant's village of origin. Most were within-district movements (76.28%; N: 865), and only a fourth were

**Table 3. Baseline sociodemographic characteristics of mobile individuals.**

| Characteristic | Iquitos, N = 396[1] | Mazan, N = 728[1] | p-value[2] |
|---|---|---|---|
| **Sex** | | | 0.044 |
| Female | 212 (54%) | 344 (47%) | |
| Male | 184 (46%) | 384 (53%) | |
| **Age** | | | <0.001 |
| < 15 | 24 (6.1%) | 196 (27%) | |
| 15 to 40 | 174 (44%) | 252 (35%) | |
| 40 to 60 | 134 (34%) | 211 (29%) | |
| 60 or older | 64 (16%) | 69 (9.5%) | |
| **Study level** | | | <0.001 |
| None or primary | 185 (47%) | 436 (71%) | |
| Secondary or higher | 211 (53%) | 182 (29%) | |
| Not available | 0 | 110 | |
| **Main economic activity** | | | 0.002 |
| Indoors | 235 (59%) | 360 (49%) | |
| Outdoors | 161 (41%) | 368 (51%) | |
| **Income in the last month** | | | <0.001 |
| Yes | 135 (34%) | 169 (23%) | |
| No | 261 (66%) | 559 (77%) | |
| Not available | 0 | 1 | |
| **Malaria diagnosis in the last 30 days** | | | 0.059 |
| No diagnosis | 378 (95%) | 710 (98%) | |
| Positive diagnosis | 18 (4.5%) | 18 (2.5%) | |
| **qPCR results** | | | <0.001 |
| Negative | 280 (76%) | 629 (94%) | |
| *P. falciparum* | 12 (3.3%) | 5 (0.7%) | |
| *P. vivax* | 69 (19%) | 38 (5.7%) | |
| Mixed infection | 6 (1.6%) | 0 (0%) | |
| Not available | 29 | 56 | |
| **Time of stay at destination** | | | 0.15 |
| Minutes | 5 (1.3%) | 13 (1.8%) | |
| Hours | 287 (72%) | 487 (67%) | |
| Days | 104 (26%) | 228 (31%) | |
| **Number of trips last month** | | | <0.001 |
| One | 168 (42%) | 489 (67%) | |
| More than one | 228 (58%) | 239 (33%) | |
| **Travel reason** | | | <0.001 |
| Economic reasons | 68 (17%) | 219 (30%) | |
| Health seeking | 11 (2.8%) | 92 (13%) | |
| Personal reasons | 142 (36%) | 89 (12%) | |
| Recreation | 92 (23%) | 83 (11%) | |
| Family matters | 20 (5.1%) | 194 (27%) | |
| Others | 63 (16%) | 51 (7.0%) | |
| **Travel destination** | | | <0.001 |
| Rural campsites | 1 (0.3%) | 31 (4.3%) | |
| Iquitos city | 290 (73%) | 100 (14%) | |
| Mazan town | 0 (0%) | 458 (63%) | |
| Others | 105 (27%) | 139 (19%) | |

(*Continued*)

**Table 3.** (Continued)

| Characteristic | Iquitos, N = 396[1] | Mazan, N = 728[1] | p-value[2] |
|---|---|---|---|
| **Overnight travel** | | | 0.4 |
| Yes | 121 (31%) | 239 (33%) | |
| No | 275 (69%) | 489 (67%) | |

[1]n (%)
[2]Pearson's Chi-squared test; Fisher's exact test

between-district travels (22.04%; N: 250). Mobile participants from Iquitos had a significantly lower within-to-between movement ratio than Mazan district participants (Iquitos ratio: 2.52; Mazan ratio: 4.23; p <0.05).

SNA of the full network (all communities included, regardless of the district) showed Libertad, Santa Rita, and Urco Miraño as the three communities with the highest degree (a sum of the in-degree and out-degree) (Fig 2 and S2 Table). Individually exploring the indices allowed us to point to Libertad as the most visited community within the screened villages (In-degree: 4, weighted in-degree: 11). Likewise, individually addressing the out-degree showed Santa Rita (Out-degree: 24, weighted out-degree: 206), Libertad (out-degree:18, weighted out-degree: 141), and Urco Mirano (Out-degree: 18, weighted out-degree: 137) as the communities with the highest number of sent travelers. In addition, we also found the ten most visited communities within the full network, including satellite destinations, non-directly assessed by our study (S3 Table). In the full network, the most visited destinations were the two largest urban centers in the assessed districts, the Iquitos city (Iquitos district, weighted in-degree: 349) and the Mazan town (Mazan district, weighted in-degree: 393), followed by local communities like 14

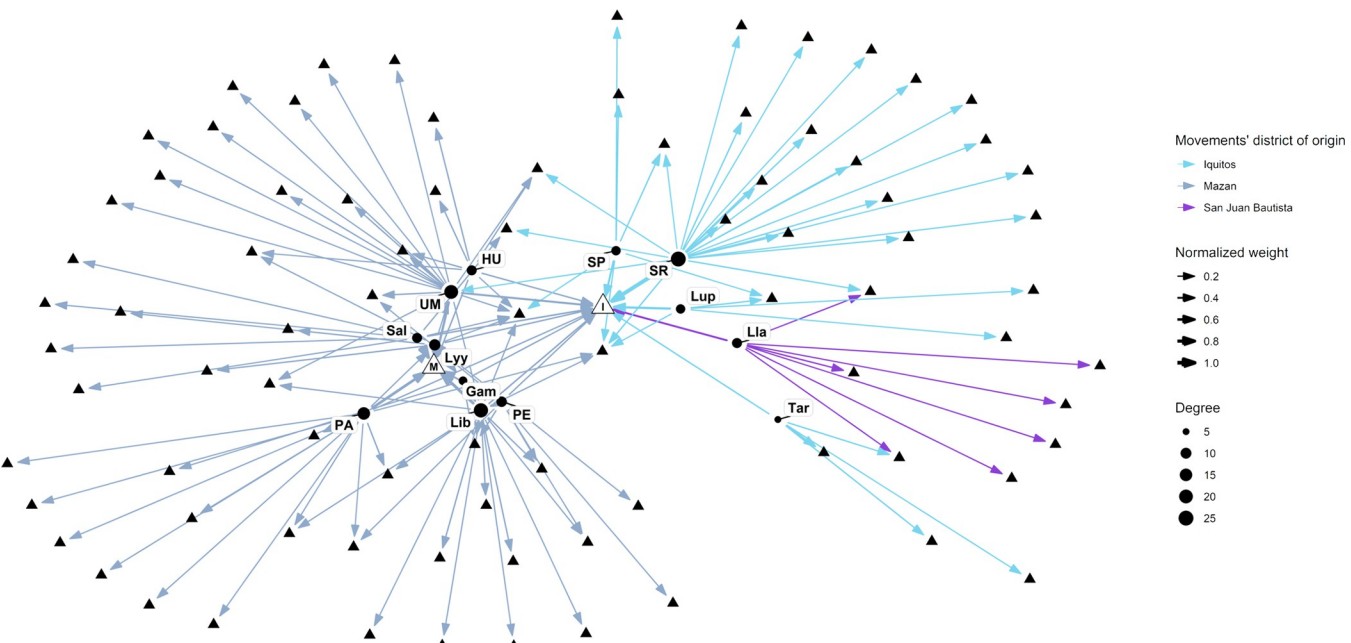

**Fig 2. Full social network analysis graph including communities from Mazan, Iquitos and San Juan Bautista districts.** This SNA graph includes communities from Iquitos, Mazan, San Jose districts. The size of the edges (arrows) varies according to the normalized number of recorded trips (weight). The size of the nodes varies according to the degree.

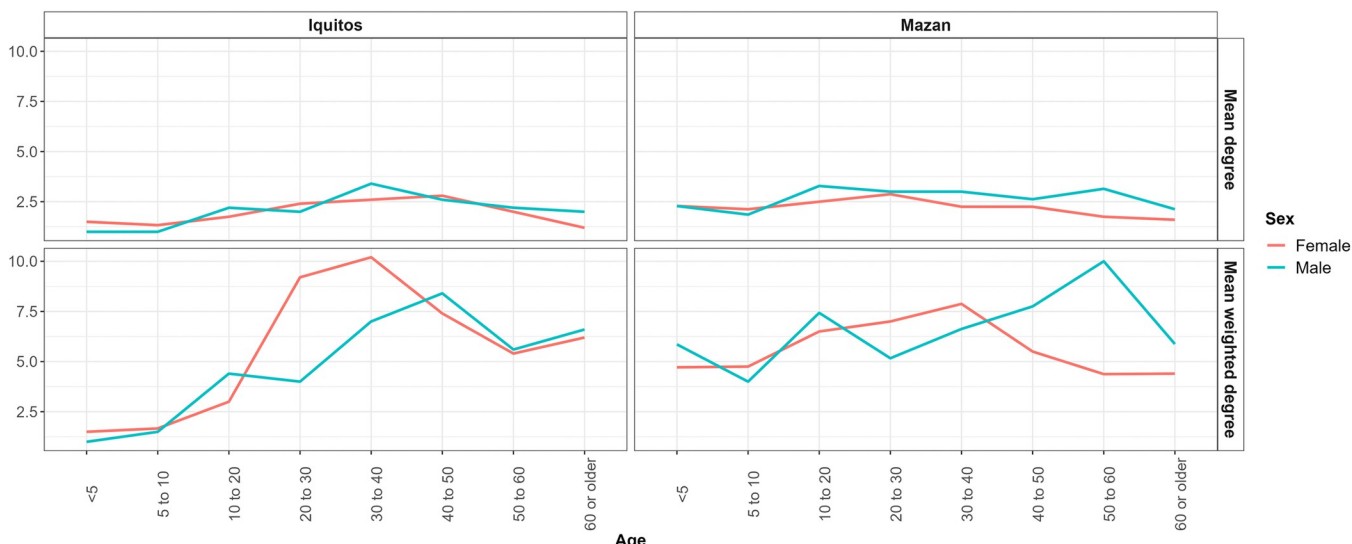

**Fig 3. Degree and weighted degree measures by district, sex and age for communities in Iquitos and Mazan districts.**

de Julio (Mazan district, weighted in-degree: 28) and Santa Clara (San JuanB district, weighted in-degree: 23). Notably, a rural campsite, where extractive activities are carried out, was among the most visited destinations (S3 Table).

The full network graph shows a clear separation between nodes according to the movements' district of origin (Fig 2). Furthermore, optimal clustering analysis using the *igraph* package in R confirms that nodes in different districts belong to different clusters. Still, multiple clusters within the same district were also present (S2 Table). Specifically, communities near Mazan town or along the Mazan River basin make up a single cluster (Libertad, Puerto Alegre, Primero de Enero, Gamitanacocha, Salvador), while distant communities make a second cluster (Urco Mirano, Huaman Urco) (Fig 2). Likewise, communities near the Iquitos city (Santa Rita, San Jose de Lupuna, and San Pedro) made the third cluster; Llanchama was included here due to shared destinations with the latter communities. A fourth cluster was also present within Iquitos, only comprising Tarapoto and its destinations.

We extended the SNA analysis following the example of Pindolia *et al* (32) by comparing the degree and weighted degree between sexes and districts in multiple age groups. We found an increase in SNA measures with age in mobile participants from the Iquitos district. Notably, such measures decreased in adults older than 40; however, the weighted degree plot showed an opposite trend in 60-year-old participants (Fig 3). These results suggest that while the number of locations visited decreased after age 40, the number of visits to those locations increased. In the Mazan district, the scenario was different. Younger participants (< 5 years) showed higher degree and weighted degree measures than their older counterparts (5 to 10 years). After age 10, degree measures showed slight variation until age 60, where a marked decrease was observed.

Interestingly, weighted degree measures showed different trends among districts, suggesting that the number of individuals visiting the same locations changed with age. Finally, exploring degree and weighted degree measures for travel reasons and age groups revealed interesting differences between districts. Participants younger than 10 years from communities in Iquitos did not travel for economic and personal reasons, while participants from Mazan communities within the same age group did (Fig 4).

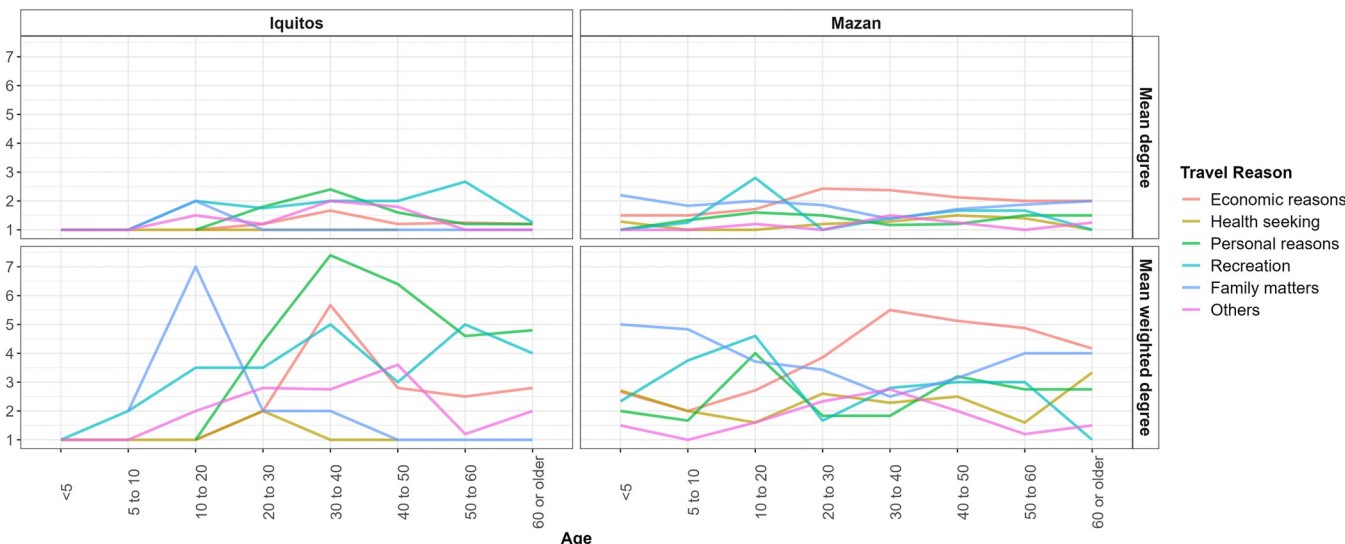

**Fig 4. Degree and weighted degree measures by district, travel reason and age for communities in Iquitos and Mazan districts.**

We complemented the analysis by exploring the distribution of income in the past month and the time of stay at destination by travel reason. We found that a large portion of participants from Iquitos who traveled due to personal (59.86%, N = 85/142) or economic reasons (29.41%, N = 20/68) received an income in the last month. Mobile participants from Mazan who reported income (34.24%, N = 75/219) traveled mostly due to economic reasons or family matters (23.83%, N = 46/193) (S1 Fig). Day-long trips for economic reasons (30.59%, N = 67/219) and family matters (23.83%, N = 46/193) were the most common among Mazan participants. Mobile individuals from Iquitos showed the highest number of day-long trips when travelling for personal reasons (14.79%, N = 21/142) and recreation (44.57%, N = 41/92) (S2 Fig).

## Assessing parasite connectivity using travel history data

We found no significant differences in the aggregated time of stay between Iquitos and Mazan (S1 Fig). Neither within nor between district movements had significantly different median aggregated times of stay. Likewise, relative importation measures at the district level were similar between Iquitos and Mazan districts. Nonetheless, the score values in Mazan were higher than in Iquitos, although not significantly different.

Aggregating travel records at the district level allowed us to assess the screened districts as sources and sinks for other evaluated districts (Fig 5 and Table 4). Iquitos communities showed higher importation values to neighboring districts like San Pablo, Fernando Lores, Parinari, Belen, and Nauta (Table 4). For communities in the Mazan district, the highest importation values were directed to the Iquitos, Requena, and Fernando Lores districts.

For Iquitos communities, the highest aggregated times of stay were shown by Llanchama, Santa Rita, and Tarapoto to locations such as Lima (the capital of the country), Iquitos city, and neighboring communities (Table 5). On the other hand, aggregated times of stay scores were led by Salvador, Urco Miraño, and Libertad in Mazan. Adjusting this measure by the incidence ratio made us lose around 61% (89 of 147) of location pairs due to non-existent malaria transmission and the inability to map location names in the DIRESA databases. Our results show higher scores from Santa Rita, Lupuna, Llanchama, and San Pedro to destinations

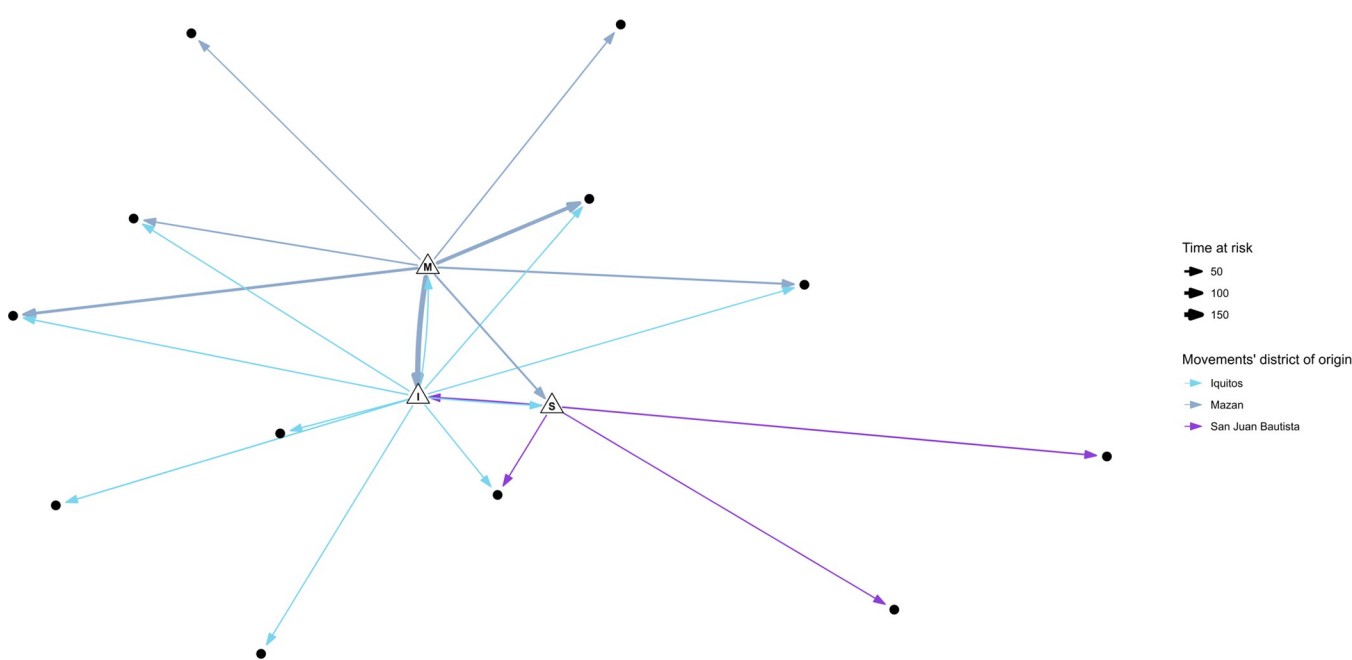

**Fig 5. Social Network Analysis graph at the district level.** Edge widthness varies according to the relative importation score.

like Iquitos city and neighboring villages (Table 6). Noteworthy, relative importation scores to Iquitos city may be overestimated due to close-to-zero transmission within the city and its effect on the mean API ratio. Similarly, scores from Mazan communities to the Iquitos city are a consequence of this overinflation. Nonetheless, neighboring communities are also within the top destinations for the score (Tables 6 and S4).

## Factors associated with *Plasmodium spp*. parasitemia in mobile individuals from Iquitos and Mazan

Our univariate analyses revealed contrasting scenarios between Iquitos and Mazan districts (Table 7). In Mazan, *Plasmodium spp*. positive qPCR diagnosis was significantly associated with male sex (OR: 1.06, 95% CI [1.02–1.10]), outdoor economic activities (OR: 1.05, 95% CI [1.01–1.09]), more than one trip in the last month (OR: 0.94, 95% CI [0.91–0.98]), and visits to rural campsites (OR: 1.15 95% CI [1.05–1.27]). Conversely, univariate analysis from Iquitos revealed associations only with health-seeking traveling (OR: 1.4, 95% CI [1.02–1.91]). These

**Table 4. Top three destinations with the highest relative importation score for each assessed district.**

| Origin | Destination | Relative importation |
|---|---|---|
| Iquitos | San pablo | 7.86 |
|  | Fernando lores | 4.59 |
|  | Nauta | 3.48 |
| Mazan | Iquitos | 179.28 |
|  | Requena | 108.69 |
|  | Fernando lores | 67.77 |
| San Juan Bautista* | Parinari | 13.76 |
|  | Iquitos | 3.9 |
|  | Belen | 1.37 |

**Table 5. Top 10 destinations with the highest aggregate time of stay for communities in Mazan and Iquitos districts.**

| District | Origin | Destination | Agg. time of stay* |
|---|---|---|---|
| Iquitos | Llanchama | Lima | 10 |
| | Santa Rita | Iquitos | 7.84 |
| | Tarapoto | Santo tomas | 4.17 |
| | Santa Rita | Lima | 3.8 |
| | Santa Rita | Cerro | 3.27 |
| | Llanchama | Pacaya Samiria | 3 |
| | Santa Rita | Santa rosa | 2.97 |
| | Santa Rita | San Pablo | 2 |
| | Llanchama | Iquitos | 1.46 |
| | Santa Rita | Chambira | 1.2 |
| Mazan | Salvador | Putumayo | 8 |
| | Salvador | Trujillo | 8 |
| | Urco Miraño | A.A.H.H. Monte Sion | 7.5 |
| | Salvador | Curaray | 7 |
| | Urco Miraño | Santa Rosa | 6.67 |
| | Urco Miraño | Iquitos | 5.32 |
| | Libertad | Paucara Urco | 5 |
| | Urco Miraño | Pucayacu | 3.5 |
| | Libertad | Quebrada Chiriaco | 3 |
| | Salvador | Iquitos | 2.95 |

*Days

**Table 6. Top 10 destinations with the highest relative importation score for communities in Iquitos and Mazan districts.**

| District | Origin | Destination | Agg. time of stay | API ratio | Relative importation |
|---|---|---|---|---|---|
| Iquitos | Santa Rita | Iquitos | 7.84 | 471 | 3691.07 |
| | Lupuna | Iquitos | 0.79 | 977.69 | 771.96 |
| | Santa Rita | San Pablo | 2 | 305.07 | 610.13 |
| | Llanchama | Iquitos | 1.46 | 270.29 | 393.8 |
| | San Pedro | Iquitos | 0.39 | 838.53 | 326.29 |
| | Lupuna | Tamshiyacu | 1 | 182.67 | 182.67 |
| | Santa Rita | Requena | 0.13 | 1,000.00 | 133.33 |
| | Santa Rita | Santa Rosa | 2.97 | 30.8 | 91.37 |
| | San Pedro | San Juan | 0.08 | 998.72 | 76.29 |
| | Tarapoto | Iquitos | 0.23 | 214.09 | 49.95 |
| Mazan | Libertad | Iquitos | 2.47 | 286.94 | 708.99 |
| | Gamitanacocha | Iquitos | 1.03 | 404.4 | 417.88 |
| | Huaman urco | Iquitos | 1.37 | 39.25 | 53.81 |
| | Salvador | Iquitos | 2.95 | 16.65 | 49.12 |
| | Lago Yuracyacu | Iquitos | 2.38 | 20.59 | 48.92 |
| | Libertad | Mazan | 2.4 | 17.02 | 40.9 |
| | Libertad | Tamshiyacu | 0.67 | 53.61 | 35.74 |
| | Libertad | Indiana | 0.17 | 201.04 | 33.51 |
| | Salvador | Requena | 0.9 | 35.35 | 31.82 |
| | Puerto Alegre | Iquitos | 0.57 | 37.17 | 21.06 |

**Table 7. Univariate analysis for *Plasmodium* spp. qPCR infection for travelers from Mazan and Iquitos.**

| Characteristic | Mazan district | | | Iquitos district | | |
|---|---|---|---|---|---|---|
| | OR | 95% CI[1] | p-value | OR | 95% CI[1] | p-value |
| **Sex** | | | | | | |
| Female | — | — | | — | — | |
| Male | 1.06 | 1.02–1.10 | 0.001 | 1.03 | 0.95–1.13 | 0.5 |
| **Age** | | | | | | |
| < 15 | — | — | | | | |
| 15 to 40 | 1.02 | 0.97–1.07 | 0.4 | — | — | |
| 40 to 60 | 1.01 | 0.96–1.07 | 0.6 | 0.99 | 0.90, 1.09 | 0.9 |
| 60 or older | 1.01 | 0.94–1.08 | 0.9 | 0.89 | 0.78, 1.00 | 0.053 |
| **Study level** | | | | | | |
| None or primary | — | — | | — | — | |
| Secondary or higher | 1 | 0.96–1.05 | 0.8 | 1.01 | 0.92, 1.10 | 0.8 |
| **Main economic activity** | | | | | | |
| Indoors | — | — | | — | — | |
| outdoors | 1.05 | 1.01–1.09 | 0.018 | 1.01 | 0.92, 1.10 | 0.8 |
| **Income in the last month** | | | | | | |
| No | — | — | | — | — | |
| Yes | 0.97 | 0.93–1.01 | 0.2 | 0.98 | 0.89, 1.07 | 0.6 |
| **House type** | | | | | | |
| Closed | — | — | | — | — | |
| Opened | 1.03 | 0.96–1.11 | 0.3 | 1.09 | 0.89, 1.32 | 0.4 |
| **Trips in the last month** | | | | | | |
| One | — | — | | — | — | |
| More than one | 0.94 | 0.91–0.98 | 0.004 | 1.04 | 0.95, 1.14 | 0.4 |
| **Overnight trip** | | | | | | |
| No | — | — | | — | — | |
| Yes | 1.03 | 0.99–1.08 | 0.1 | 0.97 | 0.88, 1.07 | 0.6 |
| **Travel destination** | | | | | | |
| Others | — | — | | — | — | |
| Rural campsites | 1.15 | 1.05–1.27 | 0.004 | 0.82 | 0.36, 1.91 | 0.7 |
| Iquitos city | 1.02 | 0.96–1.10 | 0.5 | 1.06 | 0.96, 1.17 | 0.2 |
| Mazan town | 1 | 0.95–1.06 | 0.9 | | | |
| **Travel Reason** | | | | | | |
| Economic reasons | — | — | | — | — | |
| Health seeking | 0.99 | 0.93–1.06 | 0.8 | 1.4 | 1.02–1.91 | 0.035 |
| Personal reasons | 1.03 | 0.97–1.09 | 0.4 | 1.1 | 0.98–1.25 | 0.12 |
| Recreation | 1.03 | 0.96–1.10 | 0.5 | 1.11 | 0.97–1.28 | 0.13 |
| Family matters | 0.98 | 0.93–1.03 | 0.3 | 1.07 | 0.83–1.38 | 0.6 |
| Others | 1 | 0.92–1.08 | >0.9 | 1.02 | 0.88–1.18 | 0.8 |

[1]CI = Confidence Intervals

significantly associated variables and factors with a p-value of at least 0.2 were selected for the multivariate model (Table 7).

Adjusted models were built using forward variable selection (Table 8). We maintained age and project variables in the multivariable model despite showing a p-value over the threshold to adjust for possible confounding and data collection differences. Logistic Multivariate

**Table 8. Multivariate analysis for *Plasmodium spp*. qPCR infection for travelers from Mazan.**

| Characteristic | OR[1] | 95% CI[1] | p-value |
|---|---|---|---|
| **Sex** | | | |
| Female | — | — | |
| Male | 3.44 | 1.63, 7.83 | 0.002 |
| **Trips in the last month** | | | |
| One | — | — | |
| More than one | 0.28 | 0.10, 0.65 | 0.006 |
| **Main economic activity** | | | |
| Indoors | — | — | |
| Outdoors | 2.22 | 0.83, 6.77 | 0.13 |
| **Income in the last month** | | | |
| No | — | — | |
| Yes | 0.46 | 0.17, 1.08 | 0.090 |
| **Travel destination** | | | |
| Others | — | — | |
| Rural campsites | 3.77 | 1.04, 13.8 | 0.041 |
| Iquitos city | 1.92 | 0.58, 6.56 | 0.3 |
| Mazan town | 1.22 | 0.49, 3.49 | 0.7 |
| **Project** | | | |
| P1 | — | — | |
| P3 | 1.57 | 0.74, 3.21 | 0.2 |
| **Age** | | | |
| $< 15$ | — | — | |
| 15 to 40 | 1.20 | 0.34, 4.10 | 0.8 |
| 40 to 60 | 0.75 | 0.19, 2.92 | 0.7 |
| 60 or older | 0.71 | 0.13, 3.34 | 0.7 |

[1]OR = Odds Ratio, CI = Confidence Interval

models from the Mazan district revealed significant associations with male sex (AOR: 3.44, 95% CI [1.63–7.83]) and traveling more than once (AOR: 0.28, 95% CI [0.10–0.65]). All predictor variables reported a low Variance of Inflation Factor (VIF), indicating no multicollinearity in the model. Only age showed a significant association in the Iquitos multivariable model; thus, no results are reported for this group.

## Discussion

The relationship between human mobility and malaria has a central role in large-scale malaria transmission and is a key driver of residual malaria endemicity that resists standard malaria control measures [1,3,5,10,14]. The riverine characteristics of the Amazon region drives human mobility for economic motivation, which generally differs from endemic malaria in Sub-Saharan Africa. Previous studies in Loreto have shown that human mobility, typically of subclinically-infected or asymptomatic parasitemics, drives connectivity between rural communities [12–14,35–38]https://www.zotero.org/google-docs/?Z2NOud. The present work extends previous work by providing more detailed SNA metrics using data from multiple communities in the Mazan and Iquitos districts. Our results proved that human mobility acts similarly in different communities, allowing for high connectivity and potential transmission routes. Furthermore, we explored multiple aspects of human mobility in rural and peri-urban

settings. Rural communities from the Mazan district showed many nearby communities, densely populated centers (Mazan and Indiana towns), and rural campsites as main destinations. On the other hand, communities from peri-urban settings (Iquitos district) were mostly connected to Iquitos city and some nearby communities. Although SNA and connectivity metrics yield no significant differences, geographic-specific characteristics must be considered when assessing their effect on malaria transmission [17,19].

We found high heterogeneity in malaria transmission in the Loreto region at the microgeographical level, with nearby communities having different transmission scenarios [16,17,19,39]. Malaria transmission in the Mazan district is more intense than in the Iquitos district [Mean API 2019–2020: 37.93 and 2.95, respectively] [18]. Nonetheless, one should be careful not to overgeneralize because community-specific characteristics should be accounted for. A study in Gamitanacocha (Mazan district) showed how movements to the Mazan town, a densely populated center, increase the malaria risk, likely driving parasite introduction in vulnerable communities upon return [38]. Although not previously addressed, Indiana–a contiguous densely populated center–may pose a similar risk. Notably, all communities from the Mazan district were densely connected to the Indiana and Mazan towns, emphasizing the need to quantify the real importation risk. In addition, movement to rural campsites was observed only in rural settings. Noteworthy, few papers addressed parasite transmission in these places but suggest them as a high risk for malaria transmission [40]. Moreover, a recent study proves the link between out-of-community works (mainly extractive activities) and recent parasite exposure, emphasizing the importance of such destinations [41]. Finally, inter-community parasite importation has yet to be addressed. Population genetics analyses point to parasite importation from Iquitos to Mazan districts at the district level but lose precision at the community level due to resolution limits of genotyping techniques [36,42].

In Iquitos district, the scenario was different. Communities were mainly connected with Iquitos city, an urban setting with low malaria transmission. Visits to rural campsites were rarely observed, with only one event out of 396. These results are consistent with previous reports about economic activities in peri-urban communities, commonly centered on agricultural jobs [19,38,43]. Still, parasite connectivity indices using travel data revealed no significant differences when compared against those from the Mazan district.

Exploring the parasite connectivity at the district level showed how communities from the Mazan district may act as a source for the Iquitos district. Nevertheless, this metric could be biased due to low transmission in Iquitos city, leading to overestimating the index. Furthermore, our results contradict reports that showed evidence of *P. vivax* parasite flux from the west of Loreto to the east [44]. Our calculation does not consider the possibility of introduction upon return, a phenomenon that may explain the difference in the results. If the main consequence of travels to a destination (i.e., Iquitos city, Mazan Town, Indiana, or others) is the passive importation of parasites, then the observed high degree of such movements could explain results from previous works.

Parasite connectivity results at the community level revealed a similar scenario. As Iquitos city is the main destination for communities in both districts, parasite importation metrics suggested it as a potential sink. Again, the index alone is not enough to assure parasite importation, as the presence of the vector is required for onward transmission. Removing Iquitos city from the list of destinations revealed nearby communities–many within the same district– as the main parasite sinks for both districts. Noteworthy, rural campsites' lack of transmission data forbids estimating parasite connectivity index, leading to underestimating their role in parasite importation.

We defined factors associated with *Plasmodium spp*. infection in the mobile population. Similar to other studies, male individuals and those working in outdoor activities showed

higher odds of *Plasmodium spp.* positive qPCR diagnosis [45,46]. Likewise, other studies suggested extractive-related trips as sources for *Plasmodium spp.* infection in the Mazan district [13,38,41]. Although not significant, traveling to rural campsites was also identified as a possible risk factor, further supporting the previous statement. Traveling out of the community more than once in the last month reduced the odds of positive qPCR malaria diagnosis. Reports from other communities in the district showed how extractive-related work activities involve long-term stays, sometimes for many weeks [12,13,38]. Thus, people traveling more than once may conduct short trips to low-exposure destinations, supporting the idea of long-term out-of-village extractive activities as risk factors for *Plasmodium spp.* infection in communities from the Mazan district.

Travel to nearby locations, including communities and work-related areas, is common in our study setting. The abundance of *Ny. darlingi* in non-populated areas is largely unknown, with some reports suggesting a high abundance of *Plasmodium* vectors [47,48]. Our study found that some of the travel destinations in rural communities were previously reported as uninhabited areas with high human biting rates (HBRs) [17,40,49,50]. While local malaria transmission is ongoing in our study setting, this does not exclude the possibility of travel-related infections. Previous work in the region suggests that travel-related malaria transmission is an important source of residual malaria [17,24,51]. We therefore aimed to characterize the mobile population, their reported travel, and assess factors associated with an increased *Plasmodium spp.* infection in this group. We do not neglect the possibility of local transmission but rather seek to complement the transmission scenario in rural and peri-urban settings.

Our work highlights multiple gaps that remain to be addressed. First, parasite connectivity metrics using travel data require further development. Relative parasite importation indices from this work are just a raw score with no units (i.e., number of parasites imported). Moreover, very low APIs–as in the Iquitos city–drive the parasite importation index to extremely high numbers (Table 6), overshadowing routes that could be key for parasite importation. For instance, removing movements to the Iquitos city shows pairs of communities–where vectors may be more abundant than in the city–as the ones with the highest indices.

Second, passively collected data could be more comprehensive, allowing for extensive malaria importation analyses. Even though the data provided by DIRESA-Loreto allowed us to calculate the mean API at the community level, travel history is not included in the routinely collected information. Still, we were able to complement this data with our cross-sectional collections. Due to limitations in the self-report, linking the destination names to the data provided by DIRESA-Loreto resulted in the loss of 60% of the location pairs during the parasite connectivity calculations. Studies in other regions used large amounts of passively collected data to estimate parasite importation due to human mobility, proving the relevance of such data for importation surveillance [3,5,33,52]. Third, the Peruvian NMCP does not provide guidelines to assess malaria importation within districts. Recent studies about human mobility and malaria propose detailed frameworks to classify within-country malaria importation. Still, they might not be applicable in Peru due to a lack of proper human mobility data collection [53–55]. Thus, malaria transmission between communities and rural campsites remains challenging for Peruvian NMCP. Many studies from the region highlight the relevance of parasite importation in small-scale scenarios like this, emphasizing the need to implement tailored measures to control malaria importation within districts in the Peruvian Amazon [12,13,36,38,41]. High-resolution genetic tools are essential to further differentiate malaria parasite importation at this geographical level. Currently, microsatellites and SNPs are not strong enough to differentiate between infections at the community level in neighboring settlements [36,42]. Most recent proposals, such as the Ampliseq technique, provide multitarget

panels for multiple purposes–including population genetic analyses–but require further development to achieve community-wise levels of differentiation [56].

Comprehensive passively collected data and close collaboration with research institutions would allow for routinely assessing parasite connectivity and flux in the Amazonian region [1,53]. Unlike in other scenarios, in the Amazonian setting difficulties accessing cellphone data limits the use of GPS personal devices [12,13]. There is an imperative need to develop precise and affordable methods to assess human mobility; until then, passively collected data remains a limited but powerful tool. Furthermore, high-resolution typing techniques could complement travel history obtained by passive surveillance and integrate both data types in one well-informative metric [33,52,57].

Key limitations of our work must be considered. Data collection goals from both projects (P1 and P3) were different. While P1 conducted scheduled visits in previously selected communities, P3 followed reactive sampling according to the eight-week slide positivity rate. Higher transmission in communities from the Iquitos district–where most of the P3 screenings were conducted–could be explained by that aspect. Furthermore, the lack of significant associations with *Plasmodium spp*. parasitemia in travelers from the Iquitos district could be due to current outbreaks in the screened communities. In addition, differences in variables such as education levels may be caused by adult-focused sampling in earlier screenings from P3.

We did not distinguish between *P. falciparum* and *P. vivax* transmission in our study, nor between relapses and reinfections of *P. vivax*. While previous works in the Peruvian amazon suggest species-specific differences in transmission, both parasites are transmitted by the same vector in our area of study [16]. Furthermore, most of the cases were caused by *P. vivax* and the observed associations in the multivariable model for MZ suggest that infections among travelers may be linked with travel events.

Cross-sectional screenings may underestimate connectivity. Our sampling methodology limited us by collecting travel information covering the previous 30 days; however, many individuals reported traveling more than once. The median number of travels was 3, with the highest being 22. Longitudinal projects with weekly collections showed how following a small subset of participants yields highly detailed mobility information, including trips to rural campsites for extractive activities [13,38,41]. Thus, future studies should use methods and techniques to catch detailed human-mobility data and consider transmission heterogeneity within the Peruvian Amazon.

## Conclusion

Our results demonstrate that malaria parasite connectivity patterns are not limited to some communities in the Peruvian Amazon but are a common consequence of internal human mobility in the region. Moreover, we extend our results showing key visited locations for possible malaria transmission, including rural campsites and neighboring communities. In addition, we found sub-populations of mobile individuals that may act as malaria transmission reservoirs. Understanding their mobility patterns and destinations that may act as infection sources is key to fighting malaria transmission in settings such as the Peruvian Amazon. Furthermore, we found vital gaps that should be addressed in future human-mobility malaria-related studies.

## Supporting information

**S1 Table. Data collection month per community within each district.**
(DOCX)

**S2 Table. Degrees and clusters per each node.** We did not have any recorded travel from satellite nodes (non assessed but recorded as destinations); thus, out degree was not computed.
(DOCX)

**S3 Table. Top ten destinations in the Overall Network (all communities), and per district network (Iquitos and Mazan).**
(DOCX)

**S4 Table. Complete list of relative importation scores for each pair of locations in Iquitos and Mazan districts.**
(DOCX)

**S1 Fig. Income in the last month stratified by travel reason and district.**
(PDF)

**S2 Fig. Time of stay stratified by travel reason and district.**
(PDF)

**S3 Fig. Comparison of the aggregate time of exposure and the relative importation indices between districts.**
(PDF)

**S1 Data. Supporting information file used for the analysis.**
(CSV)

## Acknowledgments

We are grateful the inhabitants in the Mazan, Iquitos, and San Juan Bautista districts for their collaboration with the project. We also thank all the Iquitos and Lima laboratories members from the Amazonian-ICEMR.

## Author Contributions

**Conceptualization:** Joaquin Gomez, Marcia C. Castro, Katherine Torres, Joseph M. Vinetz, Dionicia Gamboa.

**Data curation:** Joaquin Gomez, Stefano Garcia Castillo, Marcia C. Castro.

**Formal analysis:** Joaquin Gomez, Alessandro Grosso, Marcia C. Castro, Joseph M. Vinetz.

**Funding acquisition:** Joseph M. Vinetz.

**Investigation:** Joaquin Gomez, Alessandro Grosso, Mitchel Guzman-Guzman, Stefano Garcia Castillo, Marcia C. Castro, Katherine Torres, Joseph M. Vinetz, Dionicia Gamboa.

**Methodology:** Joaquin Gomez, Alessandro Grosso, Marcia C. Castro, Katherine Torres, Joseph M. Vinetz, Dionicia Gamboa.

**Project administration:** Mitchel Guzman-Guzman, Marcia C. Castro, Katherine Torres, Joseph M. Vinetz, Dionicia Gamboa.

**Resources:** Marcia C. Castro, Joseph M. Vinetz.

**Supervision:** Dionicia Gamboa.

**Validation:** Joseph M. Vinetz.

**Visualization:** Stefano Garcia Castillo.

**Writing – original draft:** Joaquin Gomez, Alessandro Grosso.

**Writing – review & editing:** Alessandro Grosso, Mitchel Guzman-Guzman, Stefano Garcia Castillo, Marcia C. Castro, Katherine Torres, Joseph M. Vinetz, Dionicia Gamboa.

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
