## [Decision Letter · Decision Letter 0]

22 Aug 2024

Dear Prof. Vinetz,

Thank you very much for submitting your manuscript "Human mobility and malaria risk in peri-urban and rural communities in the Peruvian Amazon" for consideration at PLOS Neglected Tropical Diseases. As with all papers reviewed by the journal, your manuscript was reviewed by members of the editorial board and by several independent reviewers. In light of the reviews (below this email), we would like to invite the resubmission of a significantly-revised version that takes into account the reviewers' comments. 

We cannot make any decision about publication until we have seen the revised manuscript and your response to the reviewers' comments. Your revised manuscript is also likely to be sent to reviewers for further evaluation.

Sincerely,

Paul J. Brindley, PhD

Editor-in-Chief

Abhay Satoskar

Section Editor

Reviewer's Responses to Questions

**Key Review Criteria Required for Acceptance?**

**Methods**

-Are the objectives of the study clearly articulated with a clear testable hypothesis stated?

-Is the study design appropriate to address the stated objectives?

-Is the population clearly described and appropriate for the hypothesis being tested?

-Is the sample size sufficient to ensure adequate power to address the hypothesis being tested?

-Were correct statistical analysis used to support conclusions?

-Are there concerns about ethical or regulatory requirements being met?

Reviewer #1: - Line 115: This section should include information about the study area. Additionally, details on the distribution of competent malaria vectors are necessary to draw conclusions about human mobility and malaria risks. If vector sources are near households, individuals may not need to travel to contract malaria.

- Explain the rationale behind selecting these three districts and the respective communities within each district.

- Line 134 and elsewhere: Verify the accuracy of the term “API is 60 cases per inhabitants.”

- Provide detailed descriptions of the data collection instruments, including their language, how they were produced, and how they were validated.

- Although the authors are using data from a series of surveys, detailed information regarding sample size estimation, inclusion and exclusion criteria, and detailed sampling procedures for each survey round should be included.

- Line 153 to 166: Clarify the reasons for including data from two projects for the Mazan district, given that the targeted population (age, etc.) differs from the other districts. Explain how these factors were controlled for in the regression models.

Reviewer #2: Minor revision

Reviewer #3: (No Response)

**Results**

-Does the analysis presented match the analysis plan?

-Are the results clearly and completely presented?

-Are the figures (Tables, Images) of sufficient quality for clarity?

Reviewer #1: - Lines 153 to 166 and Supplementary Table 1: Clarify the distinction between projects P1 and P3. P3 was conducted in all three districts among individuals over 18 years old, while P1 was only in Mazan, focusing on individuals older than three months. However, Table 2 and Table 3 include many individuals under 15 in Iquitos. Explain the categorization of age groups, particularly 15 to 40 instead of 18 to 40.

- Clarify how data for certain independent variables, such as main economic activity, income in the last month, and travel reasons, were collected for young children. It seems likely that caregivers, who were also study participants, provided this information, which may affect the distribution of data in subsequent analyses. This issue also applies to results in Tables 7 and 8, where including children in variables such as education level, main economic activities (indoor or outdoor), incomes, and travel-related information could impact the associations in the regression models.

- In Tables 2 and 3, for qPCR results, were any species other than P. falciparum and P. vivax or any unidentified species detected? What does "not applicable" mean in this context?

- In Tables 7 and 8, why did the authors combine all malaria species as Plasmodium species? P. falciparum and P. vivax have different characteristics, including relapse in P. vivax, which is not associated with travel.

- Lines 419 to 426 and 431 to 437: The authors reported OR values for some variables in opposite directions, leading to incorrect interpretations in the results, discussion, and conclusions. For example, in Table 7 for Mazan district, travel in the last month is negatively associated with Plasmodium species infection, similar to income in the last month. Additionally, in Iquitos, health-seeking related travel is associated with malaria infection. The authors should consider these variables to avoid misleading results and recommendations.

- Line 436: Avoid using the word "surprisingly."

- I noticed the authors performed a series of SNA at different stages. However, I could not find any statistical tests or SNA confirming the direct linkage between human mobility and malaria infection (even in Fig 5). I am not a demography expert, so please clarify this point.

Reviewer #2: Minor revision

Reviewer #3: (No Response)

**Conclusions**

-Are the conclusions supported by the data presented?

-Are the limitations of analysis clearly described?

-Do the authors discuss how these data can be helpful to advance our understanding of the topic under study?

-Is public health relevance addressed?

Reviewer #1: (No Response)

Reviewer #2: 

Reviewer #3: (No Response)

**Editorial and Data Presentation Modifications?**

Reviewer #1: (No Response)

Reviewer #2: Accept

Reviewer #3: (No Response)

1) Result: Social Network Analysis

It would be nice if you could provide an additional description of why comparing age against sex and travel reasons (Figures 3 and 4). I noticed that this analysis was inspired by a previous work (line 352). Is there any relation to that? If not, other demographics provided in Tables 2 and 3 can be used and compared, such as travel reasons against income or time of stay against economic activity.

2) Method: Univariate Analysis

Lines 247 and 248 mention a threshold of p-value 0.2. Please elaborate on why p-value 0.2 is used since the traditional p-value is 0.05.

3) Method: Social Network and Univariate Analysis

Could SNA and the univariate analysis be connected? It is pretty unclear to me whether the univariate (or multivariate) analysis is conducted to find significant variables before SNA or is conducted separately from SNA.

**Summary and General Comments**

Reviewer #1: Abstract

- Please include the main objective of the study.

- It would be beneficial to add specific numbers, percentages, or values from regression models in the results section to make the abstract more comprehensive.

Introduction

- The study aims to explore not only human mobility, but also other individual characteristics related to malaria risk. However, the authors have not mentioned these additional risk factors supported by other relevant studies or what new insights this study provides.

- There were some typos.

Reviewer #2: This is a really interesting and important analysis using a novel SNA approach to understand key drivers in P. vivax transmission related to the poorly understood and usually complex impacts of human mobility. 

The analysis is robust and the results well presented. Overall the analysis would have been considerably strengthened by trying to link up parasite connectivity with the rich travel history data and SNA approach alongside an analysis of parasite genetic markers. However the authors have appropriately acknowledged limitations of current panels for close proximity community-level differentiation and also clearly mentioned future applicability of larger highly multiplexed panels. 

Otherwise I have only minor comments:

Methods:

- More details in methods around the conduct of the multiple surveys and how data were combined is required (in addition to the useful supplementary figure on timing/frequency). For example, were the same households / individuals included at the multiple surveys conducted and if yes how was this encompassed in the results - were they regarded as separate individual data, or was data censored/excluded on subsequent surveys, or was there any paired/longitudinal data analysis? For any individual longitudinal data did travel reports vary over time i.e. seasonality for occupational or recreational activities which may impact results? 

- It is a bit clunky although justifiable to combine cross-sectional data from 8 adult surveys (through the P3 project) across 3 districts, and child/adult surveys (through P1) from 1 district. Was there overlap in the adult populations in Mazan district between surveys?

- Were the exact same questions asked btw P1 and P3 surveys? i.e. harmonised (states similar structure/questions in line 180 but were differences significant enough to impact results).

- Was the background API for San Juan comparable with Iquitos to allow this merging in the analysis? 

- The weighting used for the SNA metrics also needs to be defined in methods. I think this is the ‘normalized number of recorded trips’ according to Fig 2 panel - good to document how this distribution was normalised. Were there any differences in outcomes based on non-weighting vs weighting? If yes would be good to clarify and discuss what these differences mean, and if not then could consider stating this and only reporting either the raw or weighted results depending on what you think is more robust. For example it looks like in the sub-group analyses in Fig 3 that weighted values produced more variability and differentiated e.g. female vs male outcomes in the 20-30 yr bracket. 

- Could also define what in-degree and out-degree are more clearly in methods for readers not familiar with this terminology/analysis and how these are interpreted e.g. the highest degree centrality relates to the frequency of nodes being visited etc.

Results:

- The demographics of those enrolled looks pretty reasonable. What proportion of the adult household population were able to be enrolled? Was there any bias here e.g. were slightly older household members or females more commonly enrolled as at home at time of survey? 

- The distribution in age groups between districts varied considerably - with children <15 yrs comprising a much higher proportion of those surveyed in Marzan (46% vs 24%), which possibly had an impact on the lower proportion with a recent travel history (24%). This may be misleading as with a more representative sample recent travel in adults may not have been lower than Iquitos? You could consider looking at a separate adult only supplementary analysis to see whether these trends held (not necessarily reporting this in detail but to justify your current overall findings). 

- What was classified as a main economic activity for children (for the indoor/outdoor variable given there is no N/A answer)? 

- There was a large proportion of those from Iquitos without qPCR results (256/939) however the trend of relatively higher malaria incidence remains robust. 

- Figure 5 probably not necessary in main text given no results were statistically significant here. Could consider putting this as a supplemental figure instead. 

Discussion:

- Lines 487-488 relating to reports of directional transmission of Pv need referencing and further justification i.e. how were the findings in these reports made and could they be reconsidered i.e. were they accurate in view of the data you present here. 

- Re factors associated with malaria positivity. Direct links with malaria prevalence and travel / human mobility still need to be considered in view of vector bionomics, prevention and access to testing/treatment. Is the same vector present across these districts (this would then effectively control for some aspects of this) and irrespective of this what may be the contribution to the spatial heterogeneity of transmission for differences in EIR, the use of preventative measures (bed nets etc) and access to testing/treatment (including radical cure) have on these analyses? 

- It is mentioned briefly in discussion that the vector is not present in Iquitos city. Were there any other individual data available on these metrics e.g. bed net use which could have been added to the uni- and multi-variate model? Results here are also more difficult to interpret given the relapsing nature of liver stage P. vivax infections which would not be due to travel related causes - possibly worth mentioning briefly. 

- The finding that frequent travel is negatively associated with malaria prevalence is interesting as is the justification in discussion that this may be attributable to people with extended duration work related travel rather than frequent travel being at higher exposure risk. Wasn’t travel duration data available and could this be analysed in the model to try and evaluate this hypothesis?

Reviewer #3: (No Response)

PLOS authors have the option to publish the peer review history of their article (what does this mean?). If published, this will include your full peer review and any attached files.

Reviewer #1: No

Reviewer #2: No

Reviewer #3: No
---

## [Editor Report · Decision Letter 1]

21 Nov 2024

Dear Prof. Vinetz,

We are pleased to inform you that your manuscript 'Human mobility and malaria risk in peri-urban and rural communities in the Peruvian Amazon' has been provisionally accepted for publication in PLOS Neglected Tropical Diseases.

Best regards,

Paul J. Brindley, PhD

Editor-in-Chief

Abhay Satoskar

Section Editor

Shaden Kamhawi

co-Editor-in-Chief

Paul Brindley

co-Editor-in-Chief

---

## [Editor Report · Acceptance letter]

27 Dec 2024

Dear Prof. Vinetz,

We are delighted to inform you that your manuscript, "Human mobility and malaria risk in peri-urban and rural communities in the Peruvian Amazon," has been formally accepted for publication in PLOS Neglected Tropical Diseases.

Best regards,

Shaden Kamhawi

co-Editor-in-Chief

Paul Brindley

co-Editor-in-Chief
